# Targeted Dual-Modal PET/SPECT-NIR Imaging: From Building Blocks and Construction Strategies to Applications

**DOI:** 10.3390/cancers14071619

**Published:** 2022-03-23

**Authors:** Syed Muhammad Usama, Sierra C. Marker, Servando Hernandez Vargas, Solmaz AghaAmiri, Sukhen C. Ghosh, Naruhiko Ikoma, Hop S. Tran Cao, Martin J. Schnermann, Ali Azhdarinia

**Affiliations:** 1Chemical Biology Laboratory, Center for Cancer Research, National Cancer Institute, Frederick, MD 21702, USA; syed.usama@nih.gov (S.M.U.); sierra.marker@nih.gov (S.C.M.); 2The Brown Foundation Institute of Molecular Medicine, McGovern Medical School, The University of Texas Health Science Center at Houston, Houston, TX 77054, USA; servando.hernandezvargas@uth.tmc.edu (S.H.V.); solmaz.aghaamiri@uth.tmc.edu (S.A.); sukhen.ghosh@uth.tmc.edu (S.C.G.); 3Department of Surgical Oncology, The University of Texas MD Anderson Cancer Center, 1515 Holcombe Blvd., Houston, TX 77030, USA; nikoma@mdanderson.org (N.I.); hstran@mdanderson.org (H.S.T.C.)

**Keywords:** dual-modal imaging, fluorescence-guided surgery, PET/SPECT imaging, heptamethine

## Abstract

**Simple Summary:**

Targeted dual-modal imaging agents for whole body PET/SPECT imaging and fluorescence-guided surgery (FGS) have significant potential to improve surgical workflow. Here we present an overview of key design considerations and fluorescent building blocks, along with potential future directions in this exciting field.

**Abstract:**

Molecular imaging is an emerging non-invasive method to qualitatively and quantitively visualize and characterize biological processes. Among the imaging modalities, PET/SPECT and near-infrared (NIR) imaging provide synergistic properties that result in deep tissue penetration and up to cell-level resolution. Dual-modal PET/SPECT-NIR agents are commonly combined with a targeting ligand (e.g., antibody or small molecule) to engage biomolecules overexpressed in cancer, thereby enabling selective multimodal visualization of primary and metastatic tumors. The use of such agents for (i) preoperative patient selection and surgical planning and (ii) intraoperative FGS could improve surgical workflow and patient outcomes. However, the development of targeted dual-modal agents is a chemical challenge and a topic of ongoing research. In this review, we define key design considerations of targeted dual-modal imaging from a topological perspective, list targeted dual-modal probes disclosed in the last decade, review recent progress in the field of NIR fluorescent probe development, and highlight future directions in this rapidly developing field.

## 1. Introduction

Nuclear medicine is an imaging specialty that uses radiolabeled contrast agents (i.e., radiotracers) to non-invasively assess biological processes. Positron emission tomography (PET) and single-photon emission computed tomography (SPECT) are nuclear imaging modalities that generate three-dimensional images of radiotracer distribution, and are widely used in oncology, cardiology, and neurology to detect and monitor disease progression [1,2]. In cancer, diagnostic radiotracers typically comprise a targeting moiety, such as a small molecule, peptide, or antibody that is preferentially taken up by tumors, and a radionuclide that emits positrons or gamma rays for PET or SPECT imaging, respectively. The tumor-specific contrast generated by such agents has motivated imaging applications outside of nuclear medicine whose success is growing. Most notably, fluorescently-labeled agents now have over a decade of clinically proven utility in the emerging field of fluorescence-guided surgery (FGS) [3,4,5,6,7,8,9,10,11]. FGS is an intraoperative optical imaging modality that visually augments the surgical field to improve the identification of small tumors, multifocal diseases, and surgical margins. The display of real-time images in the operating room would address the limitations of existing intraoperative imaging techniques and has the potential to enable more complete tumor resections with minimal damage to normal structures (i.e., healthy tissue, nerves, and vasculature).

A conventional tumor-specific FGS agent combines a targeting component and a dye that preferably emits fluorescence in the near-infrared (NIR) spectral range (wavelengths > 700 nm), where tissue autofluorescence is low and increased depth of detection is possible [12]. Given the comparable detection sensitivities of optical and nuclear imaging (i.e., high fM–pM), there have also been extensive efforts to synthesize dual-modal FGS agents that contain both fluorescent and radioactive labels. Such agent design would broaden the imaging utility of a single agent for preoperative and intraoperative purposes (Figure 1a), while affording tools to overcome limitations of the individual modalities. For instance, fluorescence imaging is inherently semi-quantitative due to the physics of the low energy photons (~1.5 eV) involved, and thus, quantitative cross-validation of fluorescence readouts is possible at the whole body scale via PET or SPECT imaging, and at the organ scale by measuring drug distribution by gamma counting [13]. Houston and coworkers introduced the first dual-modal NIR agent using an α_v_β_3_-targeted peptide labeled with ^111^In via the chelating agent diethylenetriaminepentaacetic acid (DTPA) for gamma scintigraphy and the cyanine dye, IR-800CW, for optical imaging of melanoma in mice [14]. A major finding from their work was the ability to obtain congruent nuclear and optical signals following administration of a trace dose. This feasibility study showed for the first time that NIR and nuclear imaging can synergize and provided a foundation for developments focused on (i) chemical design strategies to simplify bioconjugation and (ii) integration of a broad range of radionuclides and dyes [15,16,17].

Several design strategies have been used to develop dual-labeled agents for nuclear/NIR imaging (detailed in Section 2). Generally, imaging scientists have combined clinically used radiometals (i.e., ^68^Ga, ^111^In, ^89^Zr) and their associated chelators, such as DTPA, 1,4,7,10-tetraazacyclododecane-1,4,7,10-tetraacetic acid (DOTA), 1,4,7-triazacyclononane-1,4,7-triacetic acid (NOTA), and desferrioxamine (DFO), with commercially available NIR dyes, such as IR-800CW, through a variety of linker technologies to biomolecules. Notably, antibody- and peptide-based approaches have pioneered the recent translation of this multimodal imaging approach and demonstrated safety and feasibility [18]. For example, sequential labeling using a validated and readily available monoclonal antibody (mAb) was implemented in the clinic with ^111^In-DOTA-girentuximab-IR-800CW for clear cell renal cell carcinoma (ccRCC, Figure 1b) resection [19]. Conversely, the use of low molecular weight agents (i.e., small molecules and peptides) typically requires more complex chemical linker strategies to preserve binding and pharmacokinetic properties. To address this challenge, ^68^Ga-NOTA-BBN-IR-800CW, which targets the gastrin-releasing peptide receptor (GRPR) using a 14 amino acid sequence peptide, was derived from a PET radiotracer and applied for glioblastoma (Figure 1c) [20]. Both of these studies illustrate the potential utility of the dual-modal approach to streamline the clinical workflow (pre- to intraoperative imaging) and mediate superior outcomes (increase surgical sensitivity).

The clinical utility of tumor-specific FGS is increasingly evident and has benefited from advances in NIR dye development. OTL38 (pafolacianine, CYTALUX^TM^) is a NIR-emitting folate-receptor targeting agent that recently gained FDA approval after demonstrating increased tumor detection in 27% of patients with ovarian cancer in a phase 3 trial (NCT03180307). A critical step in the translation of this agent was the implementation of the customized NIR dye S0456, which retained the brightness of commercially available cyanine dyes and avoided the prominent formation of side-products [21]. Several recent reviews have comprehensively described developments in chelator chemistry, and the preclinical and clinical development of dual-modal agents [18,22,23,24,25,26,27,28,29]. Here, we highlight recent advances in the development of dual-modal imaging agents bearing NIR and short-wavelength infrared (SWIR)/NIR-II (absorbance maxima > 700 nm) fluorophores. Furthermore, we focus on the fluorophore component due to extensive recent progress in this area and the potential for these agents to drive future developments. We hope that presenting these studies stimulates efforts to develop the next-generation of dual-modal PET/SPECT-NIR imaging agents with superior performance for clinical use.

## 2. Design of Dual-Modal Imaging Probes

From a topological perspective, targeted bifunctional probes may be divided into four broad categories (Figure 2). For class I, the fluorophore acts as the bifunctional component connecting the targeting ligand and radionuclide. In class II, a chelator or metal-binding ligand connects the other two functionalities. In class III, the targeting ligand is the linking element. Lastly, class IV probes are connected by a trifunctional linker. To prepare these multi-component agents, a variety of biorthogonal reactions are used in combination with conventional amide-bond forming reactions. These methods have been reviewed extensively elsewhere [30,31,32]. In this section, we include recent examples of dual-modal probe construction.

Class I strategies require a bifunctional fluorescent probe. One such example is the bifunctional cyanine fluorophore, which uses clickable handles to attach a cRGD (cyclic arginine-glycine-aspartic acid)-targeting peptide and ^18^F as the PET-emitting isotope via a one-pot synthetic method (Figure 2) [33]. The ^18^F–^19^F radiolabeling is performed on a zwitterionic organotrifluoroborate using isotope exchange (IEX) under acidic conditions. The labeling reaction results in moderate radiochemical yields (~20–25%) following purification using a C18 cartridge [34]. Tumor-to-muscle ratios of 3 using both nuclear- and fluorescence-based imaging in mice bearing glioma xenografts indicated the suitability of the NIR scaffold for dual-modal probe development. While not involving a NIR fluorophore, work by Ting and coworkers developed a related prostate-specific membrane antigen (PSMA) inhibitor, the ACUPA-Cy3-BF_3_ probe, which is currently being investigated in clinical studies for the treatment of prostate cancer [35,36,37].

Just as the fluorophore can be bifunctional, so too can the chelator element in class II strategies (Figure 2). A notable approach in this class was reported by our group, which was enabled by developing a DOTA-based analog (referred to as a multimodality chelator, MMC) with a customized pendant arm for facile bioconjugation. Our chemical design strategy maximizes the distance between the dye and targeting ligand (i.e., reduces steric interactions between these components) while maintaining the chemical footprint of low molecular weight radiotracers in the clinic. We utilized copper-free click chemistry to attach IR-800CW to the MMC-conjugated somatostatin analog MMC-TOC (TOC = Tyr^3^-octreotide) [38]. Radiolabeling with ^64^Cu allowed whole-body PET imaging in mice bearing somatostatin receptor subtype 2 (SSTR2) positive AR42J tumors. Ex vivo analysis of tumors and normal organs using gamma counting and optical imaging illustrated consistent signal localization. In subsequent studies, we modified a pendant arm of the MMC to enable stable chelation of ^68^Ga and ^67^Ga, which permitted PET imaging and delayed (up to 48 h) biodistribution analysis, respectively [39,40].

The majority of dual-modal conjugates fall into class III, where the targeting agent serves as the bifunctional group (Figure 2). These agents include dual-labeled mAbs such as ^111^In-DOTA-girentuximab-IR-800CW, as described above in Figure 1b. Using a similar topology, a pre-targeting approach was reported for detecting prostate cancer by appending DyLight 800 and DOTA through a dual-labeled diHSG (histamine–succinyl–glycine) peptide (RDC018) [41]. Mice bearing trophoblast cell surface antigen 2 (TROP-2)-expressing PC-3 prostate tumors were injected with anti-TROP-2 x anti-HSG bispecific antibody (TF12), followed by a second injection with the ^111^In-labeled DOTA-DyLight 800 conjugate. Pre-targeting led to selective uptake of RDC018 in prostate tumors for SPECT and optical imaging, with high tumor-to-background ratios (TBRs) as early as 2 h post-injection. The conjugate showed good tumor uptake in both soft tissue and bone metastases, but did exhibit high kidney accumulation, a typical issue with small molecule and peptide conjugates.

Lastly, class IV utilizes careful linker design to attach the three components (Figure 2). As described above (Figure 1c), a clinically applied class IV dual-conjugate was developed by attaching NOTA (for ^68^Ga labeling) and bombesin via a bifunctional linker to IR-800CW [20]. This study illustrated the benefits of a dual-modal agent in a clinical setting. Importantly, this method allowed for complete resection of these tumors from six of the eight patients, with the major limitation being detection of deep-seated tumors. In a separate preclinical study, ZW800-1 was linked to the chelator Dfo for ^89^Zr radiolabeling and the targeting agent cRGD (α_V_β_3_ integrin) [42]. Sibinga Mulder and coworkers showed that the conjugate could detect tumors through PET imaging. The conjugate also showed high signal-to-background ratios (SBRs) by optical imaging at early time points due to the rapid renal clearance of the conjugate.

In Table 1 and Table 2, we provide a systematic overview of all studies reported between 2011 and 2022 of PET/SPECT-NIR fluorophore (absorbance maxima > 700 nm) dual-modal imaging agents. One clear trend is that mAb-targeting agents generally use class III “double-labeling” strategies. One emerging area is the development of homogenous protein labeling methods, which has been investigated for both pretargeting and direct dual labeling [43,44,45]. Going forward, there is significant potential to apply class I, II, and IV strategies to antibody and protein targeting agents, which may also benefit from homogenous bioconjugation approaches.

## 3. Recent Developments in NIR and SWIR Fluorophores

Of the two key components of these dual-modal agents, the fluorophore and radionuclide, the latter has been reviewed extensively, including in this volume [82,83,84]. Consequently, here we focus on the fluorophore component. Fluorescence-based methods enable many cellular and in vitro biomedical experiments. In the last two decades, optical methods have been employed for in vivo applications, including in various preclinical contexts and FGS applications. In vivo imaging puts demands on these fluorescent probes that in vitro cellular imaging does not. There are significant advantages to extending the excitation and emission wavelengths into the NIR region (700–950 nm) and short-wavelength IR region (NIR-II/SWIR, 950–1700 nm) [85,86]. These advantages include enhanced penetration due to reduced absorption and scattering by tissues and blood, and reduced background signal due to lower tissue autofluorescence [87,88]. Additionally, these wavelengths reduce phototoxicity, which allows higher irradiation intensities to be used [89,90,91]. However, the chemistry of NIR probes is a significant challenge. In particular, the requirement of an extended π-system introduces a highly hydrophobic element into the chemical structure. This component must be accommodated through the addition of polar functional groups to improve water solubility and reduce probe aggregation.

### 3.1. NIR Fluorophores

Heptamethine indocyanines are privileged scaffolds for optical imaging in the NIR region [92,93]. These molecules exhibit absorbance/emission maxima of around 800 nm, exceptionally high extinction coefficients (ε ≈ 150,000–300,000 M^−1^ cm^−1^), and useful quantum yields (Φ_F_). The potential of these probes was first realized with indocyanine green (ICG), which was developed by Kodak Research Laboratories (Figure 3a). ICG was FDA approved in 1959, and was originally used to test hepatic function [94]. Due to its regulatory approval, ICG has been used in a variety of settings. In particular, the ability of ICG to target a multitude of tumor types through passive targeting mechanisms, including various liver cancers, has led to its broad use by groups around the world [95,96,97,98,99]. While useful for certain imaging applications, ICG is not suitable for active targeting. A critical advance in this area was the development of the “Cy-dyes” by Waggoneer and coworkers, which involved sulfonating the parent indocyanine indolenine rings (GE-Cy7) [100,101]. Complementing these efforts, Strekowsk and coworkers discovered that readily accessible 4’-chloro ring-modified cyanines can be converted to C4’-phenol-substituted cyanines (Figure 3a) [102]. Subsequent efforts using these modification concepts led to the broadly used commercial probes IR-800CW and DyLight 800. In particular, IR-800CW has been employed in numerous clinical trials [103,104,105]. The folate-receptor targeting OTL38, which was recently FDA approved for identifying ovarian cancer lesions, also uses this ring modification strategy [5,21,106,107,108].

Several studies have shown that persulfonated anionic probes reduce the in vivo targeting of the parent agent [109,110,111]. Moreover, it has become clear that highly charged but overall net neutral (or nearly neutral) probes offer significant benefits [109,112]. The zwitterionic fluorophore ZW800-1 was developed by the combined efforts of Choi and Henary, and Frangioni and coworkers [113,114]. While their efforts improved the properties of peptide conjugates, these probes were not optimized for mAb labeling. Through iterative efforts, our group examined a series of C4′*-O*-alkyl derivatives [115,116,117,118]. These efforts led to FNIR-Tag, a sulfonated, pegylated probe that contains a quaternary amine at the C4’ position (Figure 3a). FNIR-Tag exhibits excellent photophysical properties in a range of contexts, including on proteins, on fluorescence resonance energy transfer (FRET)-pairs, and on virus-like particles (VLPs) [119,120]. Additionally, efforts by Smith and coworkers led to the discovery of s775z (Figure 3a), a zwitterionic, pegylated probe, which was developed by introducing 2,5-disubstituted aryl derivatives at the C4’ position. This probe decreases fluorophore aggregation and improves the in vivo properties of mAb conjugates [121].

In addition to progress with actively targeted probes, there have been developments in probes with intrinsic tumor targeting properties. The C4’-chloro-containing cyanine named MHI-148 (Figure 3a) specifically localizes and persists for days in several preclinical solid tumor models (i.e., kidney, lung, brain, breast, and prostate) [122,123,124]. This targeting property has been attributed to organic anion transporting polypeptide (OATP) transporters [125,126,127,128]. A more hydrophilic version, with a quaternary amine in the cyclohexyl core, was reported by Burgess and coworkers (QuatCy) [129]. This compound maintains efficient tumor targeting in a mouse model of pancreatic cancer (NIT-91). It was also shown to covalently interact with serum proteins—a potential mechanism for other tumor-targeting C4’-chloro-substituted cyanines.

### 3.2. SWIR/NIR-II Fluorophores

Advances in instrumentation, particularly the sensitivity of InGaAs detectors, have made it possible to image in the SWIR region [130]. These wavelengths have significant potential to provide improved resolution for in vivo imaging. While efforts have been applied to generate quantum dots, carbon nanotubes, earth-doped nanoparticles, and other nanomaterial/polymer materials, active-targeting approaches with these agents come with significant challenges [131,132,133,134]. Early efforts to create organic fluorophores in this wavelength range mainly focused on exploiting benzobis(thiadiazole) (BBT, donor–acceptor–donor (DAD)) moieties [135]. However, the early versions of these probes were prone to aggregation and fluorescence quenching. To overcome this challenge, the Dai group introduced an alkoxy shielding unit and carefully optimized the chromophore, which significantly improved the properties of these probes in aqueous solutions (IR-BGP6, Figure 3b) [136,137].

Cyanines have significant potential for use in SWIR imaging. Studies by Bruns and coworkers revealed that ICG, when excited at its λ_max_, exhibits an SWIR signal at a wavelength of 1150 or 1300 nm [138]. This approach has been validated by several groups, and even applied in clinical settings. Building on these observations, various studies have sought to capitalize on the potential of cyanine derivatives in this setting. Lan and coworkers employed a thiopyrylium moiety on the monomethine to extend the photophysical properties into the SWIR region, which was targeted with the FDA-approved mAb cetuximab. Other promising efforts in this area include studies that employ flavylium heterocycles on the heptamethine cyanine scaffold. These probes have been employed for various untargeted applications and have the potential to be extended to targeted imaging [139,140,141]. We recently demonstrated that dichloro-substituted nonamethine cyanine derivatives could be modified with either catechols or through direct aryl fusion to generate FNIR-866 and FNIR-1072, respectively (Figure 3b). These probes can be used as mAb-targeted agents for multicolor surgical applications [142]. Overall, the rapid progress in the field of long-wavelength probe chemistry is opening up a range of exciting possibilities.

## 4. Future Directions of Dual-Modal Imaging

Dual-modal imaging is an emerging field that can bridge gaps in nuclear medicine and surgical oncology. There are a range of exciting unexplored directions in the design and application of these dual-modal agents. Below we focus on potential research utilizing novel fluorophore elements and methods for developing new dual-modal probes, and the applications of dual-modal imaging agents to therapeutic development.

### 4.1. Probe Development

In addition to clinical applications, dual labeling also plays an important role in characterizing the performance of FGS agents in the preclinical setting, both in vitro and in vivo, as shown in the characterization of our dual-labeled somatostatin analog [38]. For instance, in parallel with conventional fluorescence-based assays, such as flow cytometry and microscopy, dual labeling with radiometals (^64^Cu, ^67^Ga, ^68^Ga) allowed quantitative assessments of cell binding with traditional radioligand studies [38,39,40]. These experiments included a clinically validated radiotracer (^68^Ga-DOTA-TOC) as a performance benchmark and provided valuable cross-validation of fluorescence findings. Furthermore, the quantitative comparison of ^68^Ga-MMC(IR800)-TOC to ^68^Ga-DOTA-TOC in vivo strengthened our understanding of off-target uptake (i.e., blood half-life) and provides a rationale for optimization with some of the dyes discussed herein. Conversely, fluorescence imaging of a dual-labeled tumor targeting agent has been used to identify the sub-cellular distributions of radiopharmaceuticals [143]. Dual-modal imaging can also be used in translational studies by ex vivo staining of human tumor tissues and subsequent fluorescent imaging and autoradiography of the tissue section, as described by Rijpkema and coworkers [144].

### 4.2. Activatable Bifunctional Probes

Most targeted bifunctional probes utilize an “always-on” fluorophore that emits fluorescence irrespective of binding to the target protein. These fluorophores have the disadvantage of needing long in vivo clearance times before attaining good SBRs, a problem that is compounded for large molecules such as antibody conjugates [145]. This issue could be overcome by using activatable or “turn-on” fluorophores, which stay in their respective off states, or non-fluorescent forms, until activated by the desired enzymes or triggers [146,147]. Such activation methods can accelerate the ability to attain meaningful image contrast [148].

One method of probe activation uses a FRET-quenched bifunctional probe targeting MMP-14, an enzyme overexpressed in glioblastoma (Figure 4a) [149]. The probe design consists of an MMP-14 substrate peptide and an MMP-14 binding peptide connected through a linker. The peptides were attached to the quencher QC-1 (LiCor Bioscience^®^, Lincoln, NE, USA), the fluorophore IR-800CW, and the chelator NOTA for PET labeling. Upon enzymatic cleavage of the quencher from IR-800CW, which is achieved in the protease-rich tumor milieu, fluorescence turn-on is observed. In vivo, the authors observed a strong correlation between the PET and NIR signal. The fluorescence signal from the probe exhibited a high TBR of 8–11 4 h post-injection that was in accordance with PET imaging using ^68^Ga and ^64^Cu.

We recently reported a non-FRET-based turn-on strategy in the NIR range. Fluorogenic heptamethine cyanine fluorophores were generated by forming norcyanine carbamates (CyBam) [150,151]. This strategy avoids the chemical complexity of a quencher group. We have shown that these probes can be conjugated to antibodies and applied to the characterization of antibody-drug conjugate linker chemistry in vivo (Figure 4b). This approach may allow conventional mAbs to achieve high-contrast optical signals shortly after systemic administration, though further optimization of the probe component is required to enable dual-modal imaging.

### 4.3. Therapeutic Approaches 

In addition to implementing new fluorophore design strategies into dual-modal agents, therapeutic translation is another possibility. The transformation of molecular-targeted tumor imaging strategies into therapeutic modalities is a long-standing goal. One method that can be used to achieve this goal is phototherapy. Phototherapy consists of photodynamic and photothermal approaches and seeks to eradicate tumors in an irradiated region [152,153]. The photosensitizer Photofrin and 5-aminolevulinic acid (5-ALA), a biosynthetic precursor of protoporphyrin IX, are two phototherapeutic agents approved by the FDA for melanoma and glioblastoma multiforme, respectively. Several other probes based on cyanine scaffolds and other fluorophores are in preclinical and clinical trials for phototherapy [152,154]. Among these photosensitizers, IR700-DX, a silicon-phthalocyanine-based dye, has shown promising therapeutic properties upon conjugation to antibodies [155,156,157,158] or to PSMA-targeting ligands [159,160] and has recently advanced in a range of clinical studies [161,162]. For dual-modal labeling, Rijpkema and coworkers developed a multi-modal PSMA-targeting agent by labeling an anti-PSMA mAb (D2B) with IR700-DX and ^111^In (Figure 4c) [155]. They performed preoperative imaging studies with SPECT/CT and NIR imaging and showed clear tumor visualization with both modalities in the subcutaneous PSMA+ xenografts, indicating applicability of the conjugate for surgical guidance. In addition, photodynamic therapy of tumor-bearing mice after injection of the agent and subsequent treatment with 100 J/cm^2^ of NIR light resulted in a significant delay in tumor growth and longer survival rates with median survival at 73 and 16 days for the treated group and control group, respectively [159,160]. These studies demonstrate the potential of IR700-DX for applications in whole-body tumor delineation, FGS-based tumor resection, and tumor ablation through phototherapy.

## 5. Conclusions

Here we provided an overview of the chemical design features of dual-modal PET/SPECT-NIR agents, with specific focus on the fluorophore components. Key aspects of these approaches include the design of the conjugation strategy (i.e., class I–IV) and selection of the individual components, particularly dyes, which can affect targeting and pharmacokinetic properties, and thus, image contrast. The relative benefits of each of these strategies and component selection remain to be fully assessed and likely will depend on the context in which they are applied. Given the existing role of nuclear imaging in preoperative assessment and surgical planning ahead of FGS procedures, novel fluorophores could be used in innovative pairings with various imaging radionuclides. Furthermore, it would be of great utility to use the same agent for (i) preoperative patient selection and surgical planning with PET or SPECT and (ii) intraoperative FGS, thereby reducing drug development costs while affording signal congruence (i.e., arising from the same origin) regardless of detection modality. Such capabilities would address the lack of biomarker (i.e., folate receptor) specificity associated with “standard” metabolic imaging findings from ^18^F-fluorodeoxyglucose (FDG)–PET and increase the accuracy of patient selection and significance of preoperative planning. Continued multidisciplinary efforts are thus essential for defining the roles of these emerging chemical strategies and broadening the impact of intraoperative imaging in cancer.

## Figures and Tables

**Figure 1 cancers-14-01619-f001:**
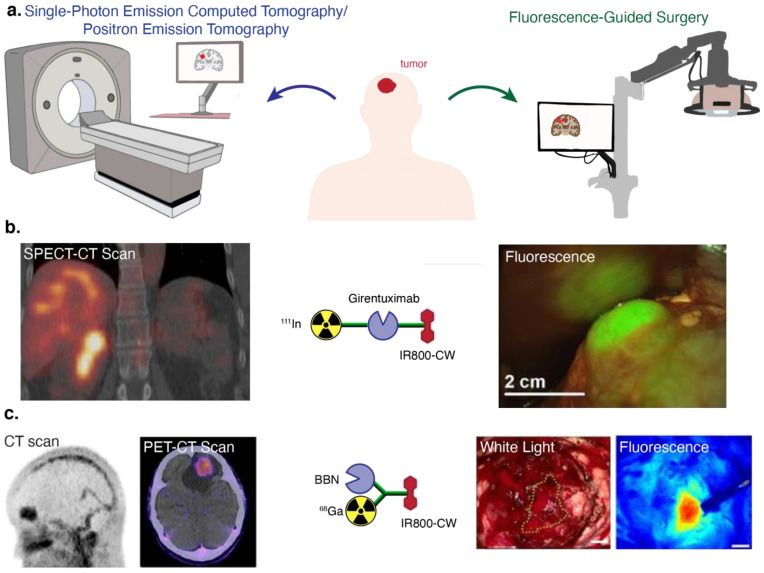
(**a**) General schematic of dual-modal imaging. (**b**) SPECT-CT and fluorescence imaging of ccRCC using ^111^In-DOTA-girentuximab-IR-800CW (girentuximab mAb targets carbonic anhydrase IX, CA-IX) [19]. (**c**) CT, PET-CT, and FGS using ^68^Ga-NOTA-BBN-IR-800CW (BBN targets GRPR) [20]. Images used according to permissions from respective journals.

**Figure 2 cancers-14-01619-f002:**
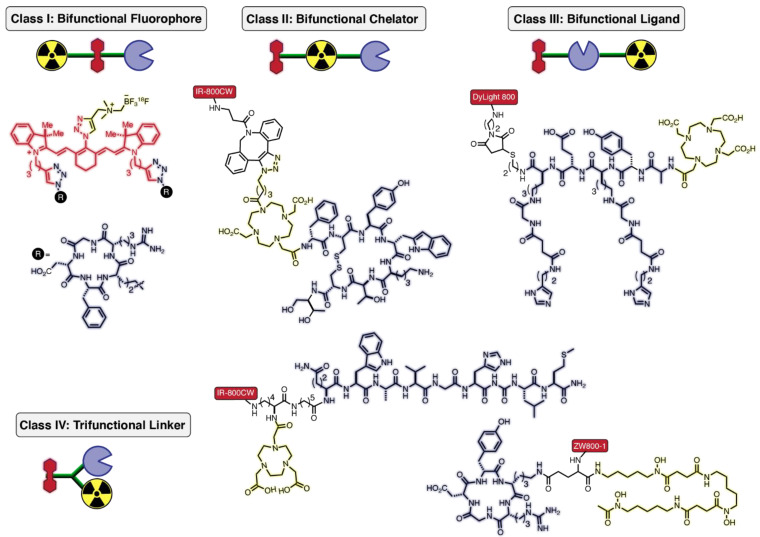
Classification of targeted-bifunctional (chelator/radionuclide and fluorophore) probes into four generic classes based on connectivity logic: class I—bifunctional fluorophores, class II—bifunctional chelators, class III—bifunctional ligands, and class IV—trifunctional linker connecting chelators, fluorophores, and targeting ligands.

**Figure 3 cancers-14-01619-f003:**
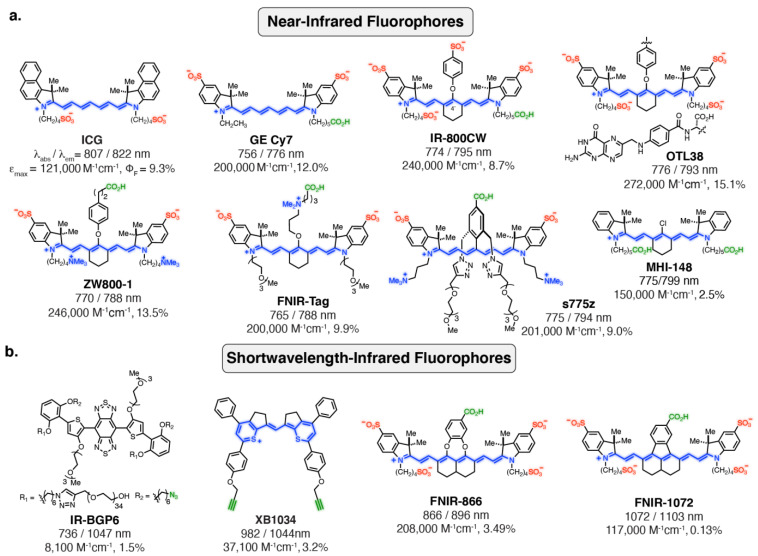
Overview of recent developments of fluorophores with absorbance/emission maxima in the (**a**) near-infrared and (**b**) short-wavelength IR regions. Blue, red, and green colors indicate cationic, anionic, and conjugable functional groups, respectively.

**Figure 4 cancers-14-01619-f004:**
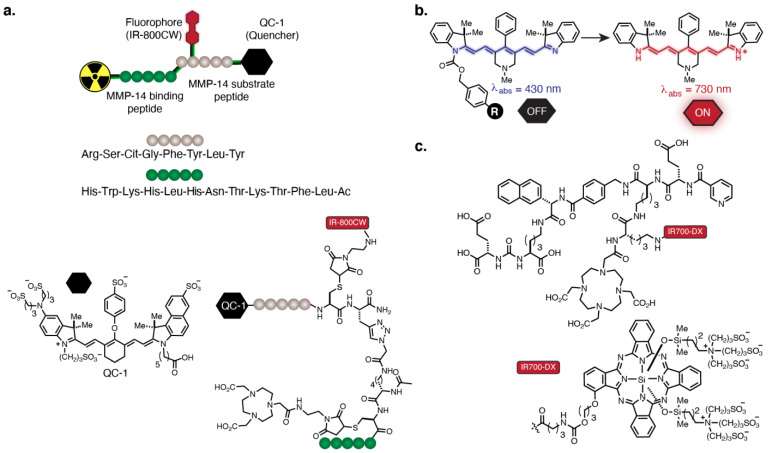
(**a**) FRET-quenched bifunctional probe targeting MMP-14. (**b**) Fluorogenic heptamethine cyanine based on self-immolative linker chemistry. (**c**) PSMA-targeted probe containing IR700-DX and ^111^In-labeled DOTA.

**Table 1 cancers-14-01619-t001:** Fluorophore–radionuclide small-molecule conjugates reported in 2011–2022.

Class	Dye	Radionuclide	Chelator	Targeting Group	Receptor	Reference
IV	IR-800CW	^68^Ga	NOTA	Bombesin	GRPR	[20]
IV	ICG Analog	^111^In	DTPA	cRGD	α_V_β_3_	[46]
III	DyLight 800	^111^In	DOTA	diHSG peptide	TF12	[41]
N/A	Sulfo-Cyanine7 and IR-800CW	^68^Ga	FSC	N/A	N/A	[47]
N/A	ICG	^68^Ga	NOTA	N/A	N/A	[48]
N/A	MHI-148	^64^Cu	DOTA	N/A	N/A	[49]
N/A	MHI-148	^68^Ga	DOTA	N/A	N/A	[50,51]
N/A	MHI-148	^99m^Tc	HYNIC	N/A	N/A	[52]
N/A	MHI-148	^64^Cu	DOTA	N/A	N/A	[53]
III	ZW800-1	^89^Zr	Dfo	cRGD	α_V_β_3_	[42]
II	IR-800CW	^64^Cu	MMC	Octreotide	Somatostatin	[38]
II	IR-800CW	^67/68^Ga	MMC	Octreotide	Somatostatin	[39,40]
IV	IR-800CW and DyLight800	^68^Ga	HBED-CC	Glu-urea-Lys(Ahx)	PSMA	[54]
III	SWIR dye	^68^Ga	DOTA	cRGD	α_V_β_3_	[55]
II	Sulfo-Cyanine7	^68^Ga	FSC	cRGD, MG11	α_V_β_3_, CCK2R	[56]
IV	IR-800CW	^111^In, ^99m^Tc	DOTA	Glu-urea-Lys analogs	PSMA	[57]

N/A = not applicable.

**Table 2 cancers-14-01619-t002:** Fluorophore-radionuclide antibody conjugates reported in 2011–2022.

Class	Dye	Radionuclide	Chelator	Targeting Group	Receptor	Reference
IV	ZW800-1(DTPA-Lys(ZW800)-Cys-NHS)structure not revealed	^111^In	DTPA	ATN-658	uPAR	[58,59]
III or IV	IR-800CW	^89^Zr	Dfo	5B1	CA19.9	[43]
III	IR-800CW	^111^In	DOTA	Farletuzumab	FRα	[60]
III	IR-800CW	^64^Cu	DOTA	MAB9601	EpCam	[61]
III	IR-800CW	^111^In	DTPA	BIWA	CD44v6	[62]
III	IR-800CW	^89^Zr	Dfo	TRC105	CD105	[63]
III	IR-800CW	^64^Cu	NOTA	TRC105	CD105	[64,65]
III	IR-800CW	^89^Zr	Dfo	TRC105, Cetuximab	CD105, EGFR	[66]
III	IR-800CW	^89^Zr	Dfo	Pertuzumab	HER2	[67]
III	IR-800CW	^89^Zr	Dfo	Cetuximab	EGFR	[68]
III	ZW800-1	^89^Zr	Dfo	YY146	CD146	[69]
III	IR-800CW	^64^Cu	NOTA	Bevacizumab	VEGF	[70]
III	IR-800CW	^111^In	DTPA, DOTA	Girentuximab	CA-IX	[19,71,72]
III	IR-800CW	^111^In	DTPA	Labetuzumab	CEA	[73]
III	IR-800CW	^111^In	DTPA	D2B	PSMA	[74]
III	IR-800CW	^111^In	DTPA	MN-14	CEA	[75]
III	IR-800CW	^111^In	DTPA	MN-14, Girentuximab, Cetuximab	CEACAM5, CA-IX, EGFR	[76]
III	IR-800CW	^64^Cu	SarAr Analog	huA33	A33	[44]
III	IR-800CW	^64^Cu	DOTA	15D3	Pgp	[77]
III	IR-800CW	^64^Cu	NOTA	CD105	TGF-β	[78]
III	XB1034	^68^Ga	NETA	Cetuximab	EGFR	[79]
IV	IR-800CW	^89^Zr	Dfo	Trastuzumab	HER2	[45]
III	IR-800CW	^124^I	N/A	A2cDb	PSCA	[80]
I	Aza-BODIPY	^111^In	DOTA	Trastuzumab	HER2	[81]

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
