# Peer review of "Targeted Dual-Modal PET/SPECT-NIR Imaging: From Building Blocks and Construction Strategies to Applications"

_cancers, 2022, doi:10.3390/cancers14071619_

Round 1

Reviewer 1 Report

This manuscript reviews the dual modal imaging probe design strategies and possible application directions. The topic is meaningful to a certain degree, however, I believe it’s not well presented and the contents is inadquate. Here are some advices:

  1. If the author specifies the “dual modal” as PET and SPECT, the basic principle of these imaging technologies should be introduced to show the authors why the agent is designed with three functional components.
  2. The title of “2. Dual modal imaging” is not appropriate, since the part actually talks about the design strategies of the probe instead of imaging.
  3. The presentation in figure 2 is confusing, the example agents here should be labeled, for example, dye part labeled with red, radionuclide with green, chelator with blue.
  4. In section 3, the classification is improper, the first type “Heptamethine Indocyanine Fluorophores” is a dye species, while, the other “SWIR” is the emission range. I suggest using the emission range, NIR-I range and NIR-II range.
  5. By “future directions”, did the author mean other applications, since 4.1 and 4.2 seemed like the new design of dual modal probes for other applications, but 4.3 didn’t mentioned design, just the photo-triggered therapy.

Author Response

please see attached doc

Reviewer 2 Report

The use of dual-modal PET/SPECT-NIR imaging agents, for both preoperative patient selection/surgical planning and intraoperative FGS, has gained considerable interest in the last decade. This short review focuses on the design of the dual-modal imaging agents and on new NIR fluorophores. The topic is definitively of interest, the manuscript is clearly written, however it lacks too many references for a review so the conclusions are not always entirely accurate.   

First, references to some important recent review papers are missing, and especially a short review from S.H. Ahn and E. Boros (Cancer Biotherapy and Radiopharmaceuticals, 2018, 33, 308-315) which addresses the same topic: “Nuclear and Optical Bimodal Imaging Probes Using Sequential Assembly: A perspective”. The review from P. Debie and S. Hernot (Frontiers in Pharmacology, 2019, 10, 510) focusing on fluorescent conjugates for FGS with a section dedicated to hybrid nuclear and fluorescent labels, also deserves to be cited. The authors should also pay attention to recent reviews on fluorophores, such as the one devoted to NIR II fluorophores (Liu et al., Chem. Rev., 2022, 122, 209-268).

When reading the sentence on page 5 line 155 (“…systematic overview of all reported studies…”) one can expect that Tables 1 and 2 effectively report all the published work in an exhaustive manner, which is not the case. For instance, in Table 1 should be added the work from Summer et al. (Bioconjugate Chem., 2017) or those of Derks et al. (Bioconjugate Chem., 2022) reporting a “class IV” PSMA targeting bimodal imaging agent. In table 2, there are also numerous studies missing, for example: Hernandez et al. (Theranostics, 2016, 16, 1918-1933), Zettlitz et al. (J. Nucl. Med., 2018, 59, 1398-1405). The work from Houghton et al. (PNAS, 2015, 112, 15850-15855) should also be reported and this example of site-specifically dually labeled antibody could be discussed in the paragraph page 5 line 132. Indeed, regarding the “class III” antibody conjugates, it could be interesting to highlight the difference between randomly conjugated compounds vs site-specifically conjugated ones.  

The statement “mAb-targeting agents invariably use class III “double -labeling” strategies” is wrong. For example, the first Ab conjugate reported in Table 2 is an example of “class IV” dual conjugate (see ref 54 and Boonstra et al., Oral Oncology, 2017). Another example of “class IV” dual conjugate has been reported by Renard et al. (Cancers, 2021, 13, 428). This work could also be cited in page 10 as it contains an IR700DX dye for PDT. The trastuzumab conjugate reported by Privat et al. (J. Med. Chem., 2021, 64, 11063-11073) represents an example of “class I” mAb dual conjugate.

These are examples with NIR dyes, and if we take into account fluorophores with emission wavelengths < 700 nm but still within the “therapeutic window”, ie between 650 and 700 nm, we can find many more examples of dual-modal PET/SPECT-NIR imaging agents of different classes, based either on antibodies, peptides or small ligands. If the authors want to limit their review to fluorophores falling within a specific range of wavelengths, this should be clearly stated. And still, regarding the construction strategy of the conjugates, the nature of the dye is not so important since one dye can be easily replaced by another one.

In conclusion, it is a well written manuscript, addressing a very interesting topic, but if the aim is to offer a comprehensive review of the literature, it needs further investigation so as not to miss representative contributions from the field.

Minor comments:

  • Page 9, paragraph 4.2. “Activatable Bifunctional Probes”. I would be less enthusiastic regarding the future of such probes in the specific context of dual-modal PET/SPECT-NIR imaging agents. Although the approach is very elegant, it is also much more demanding and the added value is not so obvious. Indeed, I agree that it could increase the SBR of the fluorescent signal but since the conjugates contain also a radioisotope for SPECT or PET imaging, it’s even more problematic to have an off-target radioactive signal. In my opinion, it is more important to work on the design of the dual conjugate to optimize its pharmacokinetics and its specificity for the target.
  • Page 9 line 289. It is a NOTA chelator and not a DOTA.
  • The example in Figure 4c is related to paragraph 4.3. and is not an example of activatable targeted bifunctional probe. Thus, it should be removed from Figure 4 and a new Figure 5 should be added.

Author Response

Please see attached doc

Reviewer 3 Report

This review of Schnermann and Azhdarinia et al. reports about the construction strategies and applications of PET/SPECT-NIR agents.

In general, the topic is highly timely and very relevant to the community. An overview about the state-of-the-art of dual-labeled PET/SPECT-NIR imaging agents is a clear benefit for the scientific community. In general, I think the authors addressed a new perspective of this topic and managed to separate their review from the recently published and very comprehensive review in Bioconjugate Chemistry (DOI: 10.1021/acs.bioconjchem.1c00503).

However, a major issue is the unclear scope and the lack of details in some chapters, especially abstract, introduction and conclusion. The authors chose a heading which suggests that this review is about PET/SPECT-NIR imaging building blocks and their application, but it is actually focused on only one building block, the NIR fluorescent probe development. The authors presume basic knowledge on PET/SPECT imaging and thus, this chapter is rarely discussed. Structures/chelators/nuclides used are not explained and some recent developments of PET/SPECT agents in combination with fluorophores are referred to other reviews. Although the authors listed many PET/SPECT-NIR agents in tables, the PET/SPECT section plays a minor role in the review. The authors themselves already point to a different focus in the introduction; I quote: “Here, we examine design considerations in the development of radioactive-NIR fluorescent probes. Particularly, we focus on the rapidly evolving field of NIR dye optimization and the growing interest in dyes for the short-wavelength infrared (SWIR) region.” (line 86-89, page 3).

Overall, the introduction and conclusion part needs more volume and structure. The authors superficially touched few examples of PET/SPECT-NIR imaging agents used in clinical studies (e.g. 111In-DOTA-girentuximab-IR-800CW) in the introduction. In my point of view, these examples are important to set off from other publications. The question raised why did the authors chose these exemplary agents showing the corresponding PET/SPECT/CT images without going into detail? There is not much information on the structure, tumour target (ccRCC???) and on the in vivo behaviour (e.g. biodistribution, accumulation,…). Furthermore, the image is somehow confusing (see listed comments and remarks below).

Nonetheless, I think that the authors have managed to focus on the fluorophore topic, especially on heptamethine indocyanine fluorophores and activatable bifunctional probes (major section, reported in heading 3 and 4). To the best of my knowledge, this has not been recently discussed in detail so far which should be the scope of this review in combination with dual-modal imaging agents (chapter 2) used in clinical studies. Moreover, shading light on the biodistribution behaviour of heptamethine dyes would be beneficial for the scope of this review as well. In addition, representative examples from original articles showing in vivo or ex vivo images, especially in chapter 3 and 4 would increase the impact as well.

In summary, I would change the focus of this review towards future directions of fluorescent dyes and optimization in combination with PET/SPECT. The authors should explain at the beginning in which directions they want to go within this review. Thus, I highly recommend the authors to re-submit an edited version of the review to the journal. The current version does not comprehensively govern the authors self-set scope.  

Further comments and remarks:

Figure 1b, page 2: First of all, 111In is commonly used as SPECT nuclide. Only PET is shown. I doubt that the image shown in figure 1b is a PET/CT scan and should be changed into SPECT/CT scan. Secondly, a brain tumour is depicted in figure 1, but the scan in 1b shows a tumour in the abdominal region (I guess it is kidney tumour). From my perspective, the image is somehow confusing due to several points, I already addressed or are mentioned below.

line 64, caption figure 1b, page 2: Please change PET and fluorescence imaging accordingly to SPECT/CT and fluorescence imaging…”.

line 64, caption figure 1, page 2: What is ccRCC and CA-IX? Please use the complete name when nowhere else explained.

line 64, caption figure 1, page 2: References are missing. Please cite the caption properly.

line 92, page 3, chapter 2: I would recommend to re-name the sub-heading of dual-modal imaging to construction strategies of PET/SPECT-NIR imaging agents or something similar.

Within the whole text, many abbreviations are used without referring to their full name or nomenclature such as the chelators DOTA, DTPA, DFO, fluorophores C-PC-NIR (line 108), ACUPA-Cy3-BF3 (line 116), QC-1 (line 288) or targets (e.g. EGFR, TF12, MMP-14, etc). In addition, it would help to show the structures of chelators and fluorophores discussed and refer to them within the text.

Line 160, table 1, page 5: What is FSC, HBED-CC? What kind of chelator is MMC? If MMC was DOTA, I would write DOTA.

line 210, page 8: I would recommend when the C4’ position is mentioned in text to refer to figure 3a.

line 213 f., page 8: Please explain once what FRET is and for what the abbreviation stands.

line 300 f., page 10: The author mentioned: ”and applied to the characterization of antibody-drug conjugate linker chemistry in vivo (Figure 4b)”. This is a highly interesting strategy. It would increase the impact to add the in vivo results of this agent to the main text and discuss the outcome in detail.

line 307, chapter 4.3, page 10: Structures of photofrin or 5-ALA may be added or the structures should be explained within the text.

line 315, page 10: Please indicate IR700-DX to figure 4c.

line 324, page 10: Is the treatment time known? If so please add.

Author Response

Please see attached doc

Reviewer 4 Report

This is an excellent, authoritative review of a rapidly developing field and I am not aware of any previous reviews with this particular focus on technical challenges.

I have no significant suggestions to improve the manuscript.

MINOR

Title.  I’m a bit worried that the terms PET/SPECT and NIR in the title may not be informative enough to a general audience. Perhaps add “radiotracer and fluorescent” with the other terms in parentheses?

Line 42. Probably should include generic name pafolacianine as well as trade name

Figure 1 contains an error in Panel B.  Indium-111 is a SPECT radionuclide rather than PET.  I believe the image is actually SPECT/CT rather than PET and seems to be taken from reference 18.

TYPOS ETC

Reference 13. The DOI did not take me to an article

The references appear to have been imported from PubMed or similar.  The isotope mass numbers are in parentheses rather than as superscripts, including refs 19, 35, 36, 37, 38, 50, 51, 61

Author Response

Please see attached doc

Round 2

Reviewer 1 Report

The authors have made proper modification, I believe the paper is acceptable with this version.

Author Response

We appreciate the kind comments

Reviewer 2 Report

Most of the concerns raised in my previous report have been addressed. Suggested references and even others have been added and the quality of the manuscript has been significantly improved.

Few remarks below:

  • Page 2 line 75. Please correct the name of NOTA: 1,4,7-triazacyclononane-1,4,7- triacetic acid.
  • Page 5 line 171. Please add “(absorbance maxima > 700 nm)” after “… NIR fluorophore”.
  • Page 6 line 174. Reference 44 may not be the best reference to cite here since it is an example of antibody fragment with site-specific conjugation of the dye but random 124I radiolabeling. I would suggest citing here, together with ref 42 and 43, reference 79 (Adumeau et al, Bioconjugate Chem 2022) which is an example of site-specific conjugation of IRDye800 + DFO chelator to a mAb, and comparison with randomly conjugated analogs.
  • Page 12 line 320. Sorry to insist, but the chelator in Figure 4a is a NOTA derivative and not a DOTA derivative. It is wrong either in the text on in Figure 4a.
  • Please read the new text carefully. There are still some typos, e.g. page 3 line 82: “…typically requires…”, line 103: “…presenting these studies stimulates efforts…”

Reviewer 3 Report

The intensive and comprehensive revison has improved the impact and scope of the review. I am convinced that the scientific community will benefit from it. Thus, I highly recommend the publication of this review in Cancers.

Author Response

We appreciate the kind comments